Cryptosporidium spp. and Giardia duodenalis emissions from humans and animals in the Three Gorges Reservoir in Chongqing, China

Huang Qian 1
Yang Ling 1
Li Bo 1
Du Huihui 1 2
Zhao Feng 1
Han Lin 1 2
Wang Qilong 1
Deng Yunjia 1
Xiao Guosheng xgs03@sanxiau.edu.cn 1 2 3
Wang Dayong 4
1 College of Biology and Food Engineering, Chongqing Three Gorges University , Wanzhou , Chongqing , China
2 Engineering Technology Research Center of Characteristic Biological Resources in Northeast Chongqing, Chongqing Three Gorges University , Wanzhou , Chongqing , China
3 Key Laboratory of Water Environment Evolution and Pollution Control in Three Gorges Reservoir, Chongqing Three Gorges University , Wanzhou , Chongqing , China
4 Medical School, Southeast University , Nanjing , Jiangsu , China
Gillespie Joseph
Electronic publication date: 2020 Nov 3
Publication date: 2020
Volume: 8
Electronic Location ID: e9985
Received 2020 Feb 25; Accepted 2020 Aug 27
Copyright: ©2020 Huang et al.
Copyright year: 2020
Copyright holder: Huang et al.
License: This is an open access article distributed under the terms of the Creative Commons Attribution License, which permits unrestricted use, distribution, reproduction and adaptation in any medium and for any purpose provided that it is properly attributed. For attribution, the original author(s), title, publication source (PeerJ) and either DOI or URL of the article must be cited.
License URL: https://creativecommons.org/licenses/by/4.0/

Keywords: Cryptosporidium spp., Giardia duodenalis, Feces, Emission, Model, Three Gorges Reservoir

Funding: Science and Technology Projects of Chongqing Municipal Education Commission, China KJQN201801225 Chongqing Technological Innovation and Application Development Project cstc2019jscx-lyjsAX0005 Chongqing Natural Science Foundation Project cstc2020jcyj-msxmX0317 This work was supported by the grant from the Science and Technology Projects of Chongqing Municipal Education Commission, China (No. KJQN201801225), the Chongqing Technological Innovation and Application Development Project (cstc2019jscx-lyjsAX0005) and the Chongqing Natural Science Foundation Project (cstc2020jcyj-msxmX0317). The funders had no role in study design, data collection and analysis, decision to publish, or preparation of the manuscript.

==============================
Cryptosporidium spp. and Giardia duodenalis are two waterborne protozoan parasites that can cause diarrhea. Human and animal feces in surface water are a major source of these pathogens. This paper presents a GloWPa-TGR-Crypto model that estimates Cryptosporidium and G. duodenalis emissions from human and animal feces in the Three Gorges Reservoir (TGR), and uses scenario analysis to predict the effects of sanitation, urbanization, and population growth on oocyst and cyst emissions for 2050. Our model estimated annual emissions of 1.6 × 1015 oocysts and 2.1 × 1015 cysts from human and animal feces, respectively. Humans were the largest contributors of oocysts and cysts, followed by pigs and poultry. Cities were hot-spots for human emissions, while districts with high livestock populations accounted for the highest animal emissions. Our model was the most sensitive to oocyst excretion rates. The results indicated that 74% and 87% of total emissions came from urban areas and humans, respectively, and 86% of total human emissions were produced by the urban population. The scenario analysis showed a potential decrease in oocyst and cyst emissions with improvements in urbanization, sanitation, wastewater treatment, and manure management, regardless of population increase. Our model can further contribute to the understanding of environmental pathways, the risk assessment of Cryptosporidium and Giardia pollution, and effective prevention and control strategies that can reduce the outbreak of waterborne diseases in the TGR and other similar watersheds.

Introduction

Cryptosporidium spp. and Giardia duodenalis are two ubiquitous parasites that can cause gastrointestinal disease in humans and many animals worldwide (Šlapeta, 2013; Ryan, Fayer & Xiao, 2014; Wu et al., 2018; Sahraoui et al., 2019). They can cause cryptosporidiosis and giardiasis, which are typically self-limiting infections in immunocompetent individuals, but are life-threatening illnesses in immunocompromised people, such as AIDS patients (Xiao & Fayer, 2008; Liu et al., 2016; Ghafari et al., 2018). In developing countries, diarrhea has been identified as the third leading cause of death (WHO, 2008), and global deaths from diarrhea are around 1.3 million annually (GBD, 2015). There are also many waterborne cryptosporidiosis and giardiasis outbreaks regularly reported in developed countries (Hoxie et al., 1997; Bartelt, Attias & Black, 2016).

Humans and many animals are important reservoirs for Cryptosporidium spp. and G. duodenalis, and large amounts of both pathogens and extremely high oocyst and cyst excretions have been traced in their feces (Graczyk & Fried, 2007; Tangtrongsup et al., 2019). Moreover, the transmission of these parasites occurs through a variety of mechanisms in the fecal-oral route, including the direct contact with or indirect ingestion of contaminated food or water (Castro-Hermida et al., 2009; Dixon, 2016; Saaed & Ongerth, 2019). These parasites can enter and pollute surface water directly through sewage sludge or indirectly through field runoff (Graczyk et al., 2008; Mons et al., 2008). Oocysts and cycsts are highly infectious, very stable in environmental water, and largely resistant to many chemical and physical inactivation agents (Carmena et al., 2007; Castro-Hermida, Gonzalez-Warleta & Mezo, 2015; Adeyemo et al., 2019), making the presence of waterborne Cryptosporidium spp. and G. duodenalis pathogens in surface water a serious public health threat (Wu et al., 2018; Li et al., 2019).

Cryptosporidiosis and giardiasis have been reported in at least 300 areas and more than 90 countries worldwide (Yang et al., 2017). Previous studies have predicted that Cryptosporidium is in 4% to 31% of the stools of immunocompetent people living in developing countries (Quihui-Cota et al., 2017) and in 1% of the stools of people with high incomes (Checkley et al., 2015). Additionally, it has been estimated that more than 200 million people are chronically infected with giardiasis, with 500,000 new cases reported each year, and that waterborne Giardia outbreaks affect approximately 10% of the world’s population (Norhayati et al., 2003; Saaed & Ongerth, 2019). To date, Cryptosporidium spp. and G. duodenalis have been found in more than 27 provincial administrative regions in China (Yang et al., 2017; Liu et al., 2020). However, there continues to be a critical lack of surveillance systems documenting and tracking protozoan infection and waterborne outbreaks in developing countries (Baldursson & Karanis, 2011; Efstratiou, Ongerth & Karanis, 2017a; Efstratiou, Ongerth & Karanis, 2017b). Cryptosporidium and Giardia have recently been added as pathogens in China’s Standards for Drinking Water Quality (GB/T5749-2006, 2007), suggesting that greater attention is being paid to waterborne parasite control in a region with no previous monitoring and reporting systems. Nevertheless, the incidence rate and risks of waterborne protozoan illness are still poorly understood in China, making it difficult to combat parasitic protozoa, manage source water, and assess future risks (An et al., 2012; Xiao et al., 2013a; Baldursson & Karanis, 2011).

The Three Gorges Reservoir (TGR), one of the world’s largest comprehensive hydropower projects, is located at the upper reaches of the Yangtze River, the longest river in Asia (He et al., 2011; Sang et al., 2019). It is an important source of water and plays a crucial role in China’s economy (Li, Huang & Qu, 2017) by optimizing their water resources. Serious pollution from agricultural activities and domestic sewage discharge have adversely affected the sustainable development of the TGR and the entire Yangtze River Basin, and pose a threat to future resources (Fu et al., 2010; Yang et al., 2015). There is a lack of observational data on Cryptosporidium and Giardia emissions from people and livestock in China. The existing data does show that monitoring programs are expensive, time-consuming, and often cannot detect or properly measure the ambient concentrations of oocysts and cysts (Efstratiou, Ongerth & Karanis, 2017a; Efstratiou, Ongerth & Karanis, 2017b; Martins et al., 2019). Understanding the environmental emissions and transmission routes of parasitic protozoa is beneficial when developing strategies to assess and mitigate waterborne diseases. In this study, we aimed to: (i) use a spatially explicit model to estimate total annual oocyst and cyst emissions from human and livestock feces in the TGR; (ii) use scenario analysis to explore the impacts of population growth, urbanization, and sanitation changes on human and animal Cryptosporidium and Giardia emissions in surface water; and (iii) contribute to a general understanding of the risk of protozoan parasites and to strategies that will control and reduce the burden of waterborne pathogens in the TGR.

Materials & Methods

Study area and model components

The TGR Area is located at 28°30′ to 31°44′N and 105°44′ to 111°39′E in the lower section of the upper reaches of the Yangtze River. It has a watershed of 46,118 km2, reaches approximately 16.8 million residents, and its river system covers 38 Chongqing districts and counties (Fig. 1). Using the area’s population (Fig. 1A) and livestock density (Fig. 1B) (Table S1), we applied the GloWPa-TGR-Crypto model to estimate oocyst and cyst emissions in the Chongqing region of the TGR (Hofstra et al., 2013). We defined an emission as the annual total number of oocysts and cysts excreted by people and livestock found in surface water. We used the emission data from 2013 for our model since the records for human and livestock populations from that year were the most complete. The model (GloWPa-TGR-Crypto) consisted of two components: a human emission model and an animal emission model. Figure 2 shows a schematic sketch of the model’s components. We identified the two types of pollution sources for the total oocysts and cysts found in the TGR. Point sources were human emissions connected to sewage systems that indirectly reached the TGR after treatment, or directly before treatment. Nonpoint sources were emissions from rural residents or livestock resulting from manure being used as fertilizer and entering surface water via runoff. Our model was partly based on Hofstra et al. (2013) and other reviews suggesting improvements in manure treatment of livestock and human emissions (Hofstra et al., 2013; Vermeulen et al., 2015; Vermeulen et al., 2017). We ran the model at both district and county levels. Our model cannot differentiate Cryptosporidium species, as there was the paucity of the prevalence and excretion rates for different species in humans and livestock.

Figure 1 Maps of population (A), livestock density (B) and river system of the Three Gorges Reservoir in Chongqing.

The data was derived from Chongqing Statistic Bureau, China (http://tjj.cq.gov.cn/).

Figure 2 Schematic sketch of the model’s components of the GloWPa-TGR-Crypto adapted from Hofstra et al. (2013).

Gray boxes are the major model subcomponents that are calculated, arrows give the pathway of Cryptosporidium and Giardia. Point sources are human emissions (H is total human emissions, Np is the human population, Op is the human average oocyst and cyst excretion rates) via sewage systems (Frem is the fraction of oocysts and cysts removed by the STP) or direct discharge into the TGR. Nonpoint sources are emissions from rural residents (without connected STP) or animals (A is total animal emissions, Na is the number of animals, Oa is the average oocyst and cyst excretion rates per animal) resulting from manure being used as fertilizer after storage and entering surface water via runoff. The oocyst and cyst emission to the TGR (E) is the main model output. Grey arrows may be important but are not taken into account in the model.

Calculating oocyst and cyst excretion in human feces (H)

Using population (P), sanitation availability (F), and excretion (Op) data from 2013, the model first estimated human oocyst and cyst emissions. We divided the human populations into four emission categories: connected sources, direct sources, diffuse sources, and non-sources. The detailed descriptions of the emission categories are provided in Kiulia et al. (2015). The model not only calculated the human emissions connected to sewers in urban and rural areas, but it also calculated direct and diffuse emissions. Unlike Kiulia et al. (2015), our model did not differentiate across age categories. In rural areas of China, a portion of fecal waste is collected and used for fertilizer and irrigation. Therefore, we assumed that human feces runoff from septic tanks and pit latrines was a diffuse source of oocysts and cysts in surface water. The two protozoan parasites’ prevalence rate was 10% in developing countries (Human Development Index (HDI) <0.785; Hofstra et al., 2013), and the average excretion rate (Op) was assumed to be 1.0 × 108 and 1.58 × 108 for oocysts and cysts, respectively (Ferguson et al., 2007; Hofstra et al., 2013). Our model used the secondary sewage treatment according to the Chinese discharge standard of pollutants for municipal wastewater treatment plants (GB18918-2002) and the Chinese technological policy for the treatment of municipal sewage and pollution control (http://www.mee.gov.cn/). The removal efficiencies were 10%, 50%, and 95%, for primary, secondary, and tertiary treatments, respectively (Hofstra et al., 2013). The results (H) were calculated using Eqs. (1) and (2):

Ki∈K1,K2,K3,K4,i∈1,2,3,4, representing four state sets.

K1—Urban connected emissions

K2—Rural connected emissions

K3—Urban direct emissions

K4—Rural diffuse emissions.

Oocyst and cyst excretions (Ki) from each human emission (i) per district were calculated as follows: (1) st.K1=CEu=Pu×Fcu×Op×1−FremK2=CEr=Pr×Fcr×Op×1−FremK3=DEu=Pu×Fdu×OpK4=DifEr=Pr×Fdifr×Op

(2) H= ∑i=14Ki

where H is the total oocyst and cyst excretion from different human emission categories in a district or county (oocysts and cycsts/year); Pu and Pr are total urban and rural populations in districts or counties, respectively; Op is the average oocyst and cyst excretion rates per person per year (oocysts and cysts/year); Fcu and Fcr are the fractions of urban and rural populations connected to a sewer, respectively; Fdu is the fraction of urban populations not connected to a sewer that is considered a direct source; Fdifr is the fraction of rural populations not using sanitation that is considered a diffuse source; and Frem is the fraction of oocysts and cysts removed by sewage treatment plants (STP). The values assumed for this study are summarized in Tables S1 and S2.

Calculating oocyst and cyst excretion in animal manure (A)

Using the number of livestock, breeding day, manure excretion, oocyst and cyst excretion rate, and prevalence rate, the model then estimated livestock oocyst and cyst emissions in 2013. We established six livestock categories: rabbits, pigs, cattle, poultry, sheep, and goats. Unlike Hofstra et al. (2013), our model used different livestock breeding day categories because each livestock species has a unique number of breeding days before slaughter and produces different amounts of manure and excretions each year. We also divided the animal populations into four emission categories: (1) connected emissions from livestock receiving manure treatment, (2) direct emissions from livestock directly discharged to surface water, (3) diffuse emissions resulting from using livestock manure as a fertilizer after storage, and (4) livestock manure that was not used for irrigation after storage or for any other use (e.g., burned for fuel) (Vermeulen et al., 2015; Vermeulen et al., 2017). We assumed that 10% of emissions would be connected emissions (Zhang et al., 2017). Illegal and undocumented direct emissions (e.g., dumped manure) were not included in our model due to a lack of data. The results (A) were calculated using Eqs. (3) and (4): Xj∈X1,X2,X3,X4,X5,j∈1,2,3,4,5

X1—Rabbit emissions

X2—Pig emissions

X3—Cattle emissions

X4—Sheep and goat emissions

X5—Poultry emissions.

We calculated oocyst and cyst excretions (Xj) from each animal species (j) per district using the following equation:

(3) Xj=Naj×Daj×Maj×Oaj×Paj

(4) A= ∑j=15Xj

where Xj is the oocyst and cyst excretions from livestock species j in a district or county (oocysts and cysts/year), Naj is the number of animals in a district or county, Daj is the breeding day for different livestock species j (days), Maj is the mean daily manure of livestock species j (kg day−1), Oaj is the oocyst and cyst excretion rate per infected livestock species j in manure (10log (oo)cysts kg−1 d−1), and Paj is the prevalence of cryptosporidiosis and giardiasis in livestock species j. The values assumed for this study are summarized in Tables S1, S3, S4, and S5.

Calculating oocyst and cyst emissions after manure storage (S)

In China, manure from human diffuse and livestock sources is collected and used for irrigation and fertilizer (Liu et al., 2019). The decay of oocysts and cysts in manure that has been stored before being applied as fertilizer in the TGR watershed is temperature-dependent during the storage period (Tang et al., 2011; Vermeulen et al., 2017). The number of oocysts and cysts in stored manure that has been loaded on land during irrigation was calculated using Eq. (5): (5) S=DifEr×Fs,h×Fv+A×Fs,a×Fv

where S is the number of oocysts and cysts in manure that has been spread on land after storage in a district or county (oocysts and cysts/year); DifEr and A are the number of oocysts and cysts in manure for rural residents (human diffuse sources) and livestock (oocysts and cysts/year), respectively; Fs,h and Fs,a are the proportions of stored manure applied as a fertilizer from rural residents and livestock, respectively (Table S2); and Fv is the proportion of average oocyst and cyst survival in the storage system.

The average oocyst and cyst survival rate (Fv) in the storage system depended on temperature (T) and storage time (ts) (Vermeulen et al., 2017). The results were calculated using Eqs. (6), (7) and (8):

(6) Ks=ln10−2.5586×T+119.63

(7) Vs=e−Ks×ts

(8) Fv=∫0tsVsdtts

where T is the average annual air temperature (°C) (Table S2), Ks is a constant based on air temperature, Vs is the survival rate of oocysts and cysts over time, and ts is the manure storage time (days) (Table S2).

Calculating oocyst and cyst runoff (R) to the TGR

Oocysts and cysts in stored manure that have been applied to agricultural land as a fertilizer are transported from land to rivers largely via surface runoff (Velthof et al., 2009). Our model estimated oocyst and cyst runoff using the amounts of manure applied as fertilizer, maximal surface runoff, and a set of reduction factors (Velthof et al., 2009; Hofstra et al., 2013). The results (R) were calculated using Eq. (9): (9) R=S×Frun,max×flu×fp×frc×fs

where R is the number of oocysts and cysts in manure applied as a fertilizer that reached the TGR via surface runoff in a district or county (oocysts and cysts/year), S is the number of oocysts and cysts in manure that was spread on land after storage (oocyst and cysts/year), Frun,max is the fraction of maximum surface runoff across different slope classes, flu is the reduction factor for land use, fp is the reduction factor for average annual precipitation, frc is the reduction factor for rock depth, and fs is the reduction factor for soil type. The values assumed for this study are summarized in Table S2.

Calculating total emissions (E) and mean concentrations (C) of oocysts and cysts

Our model defined total oocyst and cyst emissions as the annual number of oocysts and cysts per district or county in the TGR. The results (E) were calculated using Eq. (10): (10) E=CEu+CEr+DEu+R

where E is the total oocyst and cyst emissions from humans and animals in a district or county (oocysts and cysts/year), CEu is oocyst and cyst emissions in the TGR by urban populations connected to STP in a district or county, CEr is oocyst and cyst emissions in the TGR by rural populations connected to STP in a district or county, DEu is direct oocyst and cyst emissions in the TGR by urban populations in a district or county, and R is oocyst and cyst emissions in the TGR from human and livestock manure that has been applied as a fertilizer via runoff in a district or county.

According to total emissions from the two protozoa in Chongqing and the TGR’s hydrological information, we preliminarily calculated mean Cryptosporidium and Giardia concentrations in the TGR in 2013 using the GloWPa-Crypto C1 model (Vermeulen et al., 2019). Mean concentrations were calculated using Eq. (11). All equations and parameters used in this study were showed in Table S11. (11) C=Et×e−KT+KR+KS×tQs

Where C is mean concentrations of oocysts and cysts (oocysts and cysts 10 L−1), Et is the sum of total oocyst and cyst emissions from humans and animals in all districts or counties in Chongqing (oocysts and cysts/year). KT, KR,  and KS represent loss rate constants of temperature, solar radiation, and sedimentation, respectively (day−1). t is residence time of oocysts and cysts in Chongqing section of the TGR, Qs is the sum of the TGR’s annual inflow and storage capacity (m3 year−1).

Sensitivity analysis

We tested our model’s sensitivity to change using input parameters in a nominal range sensitivity analysis (NRSA). Input parameter values were based on reasonable lower and upper ranges of a base model, and we tested each variable individually (Vermeulen et al., 2015; Vermeulen et al., 2017). We selected the NRSA because it provides quantitative insight into the individual impact of different parameters on the model’s outcome. Tables S6–S8 present the sensitivity analysis input variables.

Predicting total oocyst and cyst emissions for 2050

To explore the impact of future population, urbanization, and sanitation changes on human and animal Cryptosporidium and Giardia emissions in the TGR, we divided the emissions into urban resident, rural resident, and livestock categories to predict the total oocyst and cyst emissions for 2050 based on three scenarios. China’s projected population, urbanization, and livestock production data for 2050 can be found in the Shared Socioeconomic Pathways (SSPs) database (Zhao, 2018; Huang et al., 2019; Chen et al., 2020) (https://tntcat.iiasa.ac.at/SspDb). We based Scenario 1 on SSP1, which is entitled “Sustainability—Taking the green road” and emphasizes sustainability, well-being, and equity. In this scenario, there is moderate population change and well-planned urbanization (O’Neill et al., 2015; Jiang & O’Neill, 2017; Zhao, 2018; Huang et al., 2019; Chen et al., 2020). Scenario 2 was based on SSP3, which is entitled “Regional rivalry—A rocky road” and emphasizes regional progress. In this scenario, China’s population change is significant and urbanization is unplanned (O’Neill et al., 2015; Jiang & O’Neill, 2017; Zhao, 2018; Huang et al., 2019; Chen et al., 2020). To emphasize the importance of wastewater and manure treatment, we created Scenario 3 as a variation of Scenario 1 based on Hofstra & Vermeulen (2016). This scenario has the same population, urbanization, and sanitation changes as Scenario 1, but with the insufficient sewage and manure treatments from 2013. Since there were no available data for individual livestock species, we assumed that all livestock species will grow by the same percentage noted in the SSPs database. We also assumed that there will be changes only in population and livestock numbers, not in any other parameters (e.g., oocyst and cyst excretion rates and prevalence) (Iqbal, Islam & Hofstra, 2019). We based our sanitation, wastewater, and manure treatment predictions for these three scenarios on previous literature reviews (Hofstra & Vermeulen, 2016; Iqbal, Islam & Hofstra, 2019). Table S9 provides an overview of the scenarios.

Results

Emissions and mean concentrations of oocysts and cysts in the TGR in 2013

Cryptosporidium oocyst and Giardia cyst emissions from humans, rabbits, pigs, cattle, sheep, goats, and poultry found in the TGR in 2013 are shown in Fig. 3. Chongqing had a total of 1.6 × 1015 oocysts/year and 2.1 × 1015 cysts/year of Cryptosporidium and Giardia emissions. Human Cryptosporidium and Giardia emissions contained a total of 1.2 × 1015 oocysts/year and 2.0 × 1015 cysts/year, and animal emissions had a total of 3.4 × 1014 oocysts/year and 1.5 × 1014 cysts/year. Humans and animals were responsible for 42% and 10% of total emissions in the TGR, respectively. Humans were responsible for 78% of oocyst emissions, followed by 14% from pigs, and 8% from poultry. Humans were responsible for 93% of cyst emissions, followed by 6% from pigs, and 0.5% from cattle. Ultimately, we found that humans were the dominant source of oocysts and cysts, followed by pigs, poultry, and cattle. The mean Cryptosporidium and Giardia concentrations in the TGR in 2013 were 22 oocysts/10 L and 28 cysts/10 L, respectively.

Figure 3 Total Cryptosporidium oocyst and Giardia cyst emissions in the TGR for Chongqing in 2013.

Oocyst and cyst emission sanitation types

We immediately observed the differences in sanitation types (connected emissions, direct emissions, diffuse emissions, and non-source) across the human, livestock, urban, and rural populations (Fig. 4). We found that 49% of the populations were connected to a sewer, 36% had diffuse sources, 13% had direct sources, and 2% were non-source. In livestock populations, only 10% were connected to a sewer (manure treatment), 80% produced diffuse emissions, and 10% were non-source. We divided the human and livestock emissions by region: urban areas (made up of urban residents) and rural areas (made up of rural residents and all livestock). In urban areas, the emissions connected to a sewer were prominent (78%), followed by direct sources (22%). In rural areas, diffuse sources produced approximately 85% of total emissions, followed by connected sources (9%), and non-source (6%).

Figure 4 Fraction (%) of four oocyst and cyst emission categories caused by the population in the model.

Four emission categories are used in our model: (1) connected emissions (the fraction of humans connecting to sewage systems or livestock receiving manure treatment), (2) direct emissions (the fraction of urban residents not connecting to sewage systems), (3) diffuse emissions (the fraction of rural residents or livestock resulting from manure being used as fertilizer after storage and entering surface water via runoff) and (4) non-source (the fraction of oocysts and cysts are not emissions to the environment).

Spatial distribution of oocyst and cyst emissions in the TGR in 2013

Our model produced a spatial distribution of Cryptosporidium and Giardia emissions in the TGR for each Chongqing district or county from 2013 (Fig. 5). The total human Cryptosporidium emissions ranged from 5.4 × 1012 to 7.4 × 1013 oocysts/district (Fig. 5A) and the total human Giardia emissions ranged from 8.5 × 1012 to 1.2 × 1014 cysts/district (Fig. 5B). Overall, the emission spatial differences depended on population density and urbanization rate. The largest emissions were from the densely-populated Yubei, Wanzhou, and Jiulongpo districts.

Figure 5 Cryptosporidium oocyst and Giardia cyst emissions in the TGR for each Chongqing district or county from 2013.

(A) Human oocyst emissions, (B) human cyst emissions, (C) animal oocyst emissions, (D) animal cyst emissions, (E) total Cryptosporidium oocyst emissions, (F) total Giardia cyst emissions.

The total animal source emissions ranged from 0 to 1.8 × 1013 oocysts/district (Fig. 5C) and 0 to 7.2 × 1013 cysts/district (Fig. 5D) for Cryptosporidium and Giardia, respectively. We based our results on the number of animals, manure production, manure treatment and runoff, and invariant oocyst and cyst emissions from each animal category over one year. The lowest emissions were observed in areas with low animal populations, such as the downtown districts of Yuzhong, Dadukou, and Nanan.

The total Cryptosporidium and Giardia emissions ranged from 1.0 × 1013 to 8.1 × 1013 oocysts/district and 1.0 × 1013 to 1.2 × 1014 cysts/district, respectively (Figs. 5E and 5F). Total human emissions were approximately six-fold higher than animal emissions and played a decisive role in total emission distribution. We found slightly more cysts than oocysts in total emissions and human emissions. In contrast, there were slightly more oocysts than cysts in animal emissions. The highest total emissions were found in areas with large animal and human populations, such as the main districts of Wanzhou, Yubei, and Hechuan.

Human Cryptosporidium and Giardia emissions from urban and rural areas can be found in Fig. 6. In urban areas, Cryptosporidium and Giardia emissions ranged from 3.5 × 1012 to 7.0 × 1013 oocysts/district and 5.5 × 1012 to 1.1 × 1014 cysts/district, respectively (Figs. 6A and 6C). In rural areas, the emissions ranged from 0 to 9.8 × 1012 oocysts/district and 0 to 1.6 × 1013 cysts/district (Figs. 6B and 6D). Rural emissions were spread over much larger areas than urban emissions. Human emissions in urban areas were six-fold higher than in rural areas and played a crucial role in total human emission distribution.

Figure 6 Human emissions of Cryptosporidium oocysts and Giardia cysts in urban and rural areas for each Chongqing district or county from 2013.

(A) Urban oocyst, (B) rural oocyst, (C) urban cyst and (D) rural cyst.

Sensitivity analysis

Since there were limited observational data on Cryptosporidium and Giardia, we performed a sensitivity analysis to verify the GloWPa-TGR-Crypto model’s performance. The sensitivity analysis (Tables S6–S8) showed that the model was the most sensitive to changes in excretion rate (shown for 1 log unit change in excretion rates), particularly the excretion rates of humans (factor 8.03), pigs (factor 2.22), and poultry (factor 1.74). The model was more sensitive to prevalence changes in humans, pigs, and poultry (factors 1.39, 1.14, and 1.08, respectively). The results confirmed that humans, pigs, and poultry were the dominant sources of oocyst and cyst emissions. Besides excretion rate and prevalence, the model was most sensitive to changes in the amount of runoff, STP oocyst and cyst removal efficiencies, the amount of connected emissions, human population, and manure storage time (factors 1.30, 1.23, 1.21, 1.16, and 1.11, respectively), as these parameters affected oocyst and cyst survival and emissions. The model was not very sensitive to changes in the amount of rural resident feces applied as fertilizer, rural wastewater treatment, and the excretion rates and prevalence of animal species that did not contribute much to the total oocyst and cyst emissions (e.g., cattle, rabbits, sheep, and goats).

Scenario analysis: the effect of population, urbanization, and sanitation changes in 2050

In Scenario 1, moderate population change, planned urbanization, and strong improvements in sanitation, wastewater, and manure treatments will decrease the total emissions in the TGR to 9.5 × 1014 oocysts/year and 1.2 × 1015 cysts/year by 2050 (Fig. 7). This would reduce approximately 40% of the Cryptosporidium emissions and 44% of the Giardia emissions measured in 2013. All emissions from all three sources would decrease, with a notable 61% decrease for rural residents (Table S10). Figures 8B, 8E, 9B and 9E show the decrease across all regions in Scenario 1. The largest decline would be found in the Yubei and Jiulongpo districts, where assumed urbanization rates would increase to 100% and 99%, respectively, and 99% of domestic sewers would obtain secondary or tertiary treatment. Scenario 1 also shows changes in the contributions to total emissions. Urban residents would be responsible for 64% and 81% of Cryptosporidium and Giardia emissions, respectively, which would be a 3% decrease and a 1% increase from 2013. Rural Cryptosporidium and Giardia emissions would decrease from 11% to 7% and 13% to 9%, respectively. Livestock Cryptosporidium and Giardia emissions would increase from 22% to 29% and 7% to 10%, respectively.

Figure 7 Total Cryptosporidium oocyst and Giardia cyst emissions from urban residents, rural residents and livestock in the Chongqing area of the TGR in 2013 and for 2050 based on three scenarios.

Figure 8 Differences of Cryptosporidium emissions between 2013 and 2050 for each Chongqing district or county based three scenarios.

Emissions in 2013 (A), emissions of scenario 1 (B), scenario 2 (C), and scenario 3 (D) for 2050, and emission differences between 2050 and 2013 for scenario 1 (E), scenario 2 (F), and scenario 3 (G).

Figure 9 Differences of Giardia emissions between 2013 and 2050 for each Chongqing district or county based three scenarios.

Emissions in 2013 (A), emissions of scenario 1 (B), scenario 2 (C), and scenario 3 (D) for 2050, and emission differences between 2050 and 2013 for scenario 1 (E), scenario 2 (F), and scenario 3 (G).

In Scenario 2, Cryptosporidium and Giardia emissions are expected to increase to 1.9 × 1015 oocysts/year and 2.4 × 1015 cysts/year by 2050 (Fig. 7), would be 19% and 12% growth, respectively, from 2013. Emissions from urban residents and livestock would increase (Table S10) due to strong population growth, unplanned urbanization, limited sanitation, and expanded livestock production practices where untreated manure used as fertilizer is emitted into surface water. Emissions from rural residents would decrease 8% because the rate of urbanization would increase while the same sanitation practices from 2013 are used by that smaller rural population. Figures 8C, 8F, 9C and 9F show that total emissions would increase in all regions (particularly the Wanzhou and Yubei districts) because of strong population growth and limited environmental regulation. In Scenario 2, urban residents would account for 63% and 79%, rural residents would account for 9% and 11%, and livestock would account for 29% and 10% of Cryptosporidium and Giardia emissions, respectively, by 2050.

Scenario 3 has the same population, urbanization, and sanitation changes as Scenario 1, but with limited wastewater and manure treatment facilities. Scenario 3 has the highest Cryptosporidium and Giardia emissions of all the scenarios. Total emissions would increase to 2.0 × 1015 oocysts/year and 2.7 × 1015 cysts/year, with 29% and 27% growth compared to 2013, respectively (Fig. 7). Livestock would see the most growth in emissions (an increase of 42%) (Table S10). Figures 8D, 8G, 9D and 9G show that an increase in emissions across all regions, except in regions with assumed urbanization rates of 100% and where 50% of emissions obtain secondary treatment (such as the Yuzhong and Shapingba districts). This result highlights the importance of wastewater and manure treatment. Connecting populations to sewers without appropriate sewage treatment introduces more waterborne pathogens to surface water, affecting water quality.

Discussion

The increase in Cryptosporidium and Giardia surface water pollution in China is traced primarily to human and animal feces. China has one of the largest amounts of Cryptosporidium emissions from feces (1016 oocysts/year; Hofstra et al., 2013; Hofstra & Vermeulen, 2016; Vermeulen et al., 2019), but Cryptosporidium and Giardia emissions from human and animal feces in surface water across different Chinese provinces or regions have not been closely studied. The TGR, developed by the China Yangtze Three Gorges Project as one of the largest freshwater resources in the world, suffers from Cryptosporidium and Giardia pollution (Xiao et al., 2013a; Xiao et al., 2013b; Liu et al., 2019). Using data from 2013, we built a GloWPa-TGR-Crypto model to estimate Cryptosporidium spp. and G. duodenalis emissions from human and livestock in the TGR. We also used scenario analyses to predict the effects of sanitation, urbanization, and population changes on oocyst and cyst emissions for 2050. Our study can be used to better understand the risk of water contamination in the TGR and to ensure that the reservoir is adequately protected and treated. This knowledge can also contribute to the implementation of the Water Pollution Control Action Plan (i.e., the Ten-point Water Plan), which was sanctioned by the Chinese government to prevent and control water pollution (Wu et al., 2016). Additionally, our results can serve as an example for other studies on important waterborne pathogens from fecal wastes and wastewater, particularly in developing countries.

Using the GloWPa-TGR-Crypto model, we estimated that the total Cryptosporidium and Giardia emissions from human and livestock feces in Chongqing in 2013 were 1.6 × 1015 oocysts/year and 2.1 × 1015 cysts/year, respectively. Using the total emissions from the two protozoa, the TGR’s hydrological information (such as water temperature, solar radiation, and river depth; Table S11), and the GloWPa-Crypto C1 model (Vermeulen et al., 2019), we preliminarily calculated the mean Cryptosporidium and Giardia concentrations in the TGR in 2013 to be 22 oocysts/10 L and 28 cysts/10 L, respectively. Xiao et al. (2013a) reported that Cryptosporidium oocysts and Giardia cysts are widely distributed in the TGR, with concentrations ranging from 0 to 28.8 oocysts/10 L for Cryptosporidium and 0 to 32.13 cysts/10 L for Giardia in the Yangtze River’s mainstream and the backwater areas of tributaries and cities. Liu et al. (2019) used a calibrated hydrological and sediment transport model to investigate the population, livestock, agriculture, and wastewater treatment plants in the Daning River watershed, a small tributary of the TGR in Chongqing, and found Cryptosporidium concentrations of 0.7–33.4 oocysts/10 L. The results from our model were similar to the results found in other studies. Because of the adsorption, deposition, inactivation, and recovery efficiencies of Cryptosporidium and Giardia in water, the oocyst and cyst concentrations in the surface waters of streams and rivers were significantly reduced (Antenucci, Brookes & Hipsey, 2005; Searcy et al., 2006; Vermeulen et al., 2019). Therefore, the validity of our model was confirmed.

Human and animal feces are main sources of Cryptosporidium and Giardia emissions in surface water. In this study, the majority of human emissions were from densely populated urban areas (Fig. 6). In those urban areas, a fraction of human emissions were not connected to sewers and sewage was not efficiently treated. We found high concentrations of Cryptosporidium (6.01–16.3 oocysts/10 L) and Giardia (59.52–88.21 cysts/10 L) in the effluent from wastewater treatment plants in the TGR area (Xiao et al., 2013a; Xiao et al., 2013b), and 16.5 × 108 tons of sewage were discharged into the TGR, mainly in urban areas (Xiao et al., 2013a). In rural areas, only 9% of the population was connected to sewage systems and a large portion of untreated rural sewage was used as potential agricultural irrigation water (Hou, Wang & Zhao, 2012). Therefore, we assumed that large amounts of raw sewage were used as a diffuse source that was dumped as a fertilizer after storage into the farmland, where it could then enter tributaries and the mainstream of the Yangtze River via runoff.

We found a lower amount of animal Cryptosporidium and Giardia emissions than human emissions because only 10% of diffuse emissions reached the TGR through runoff. Unlike the original model created by Hofstra et al. (2013), which estimated livestock oocyst and cyst emissions in surface water, we assumed that a portion of the manure received treatment during storage and before it was applied to soil (An et al., 2017). Recent studies also reported that oocyst emissions on land were associated with mesophilic or thermophilic anaerobic digestion during manure treatment, and could be reduced by several log units (Hutchison et al., 2005; Vermeulen et al., 2017). Additionally, animal emissions are still important. The total global Cryptosporidium emissions from livestock manure are up to 3.2 × 1023 oocysts/year (Vermeulen et al., 2017). In 2010, China had a total of 1.9 billion tons of livestock manure, 227 million tons of livestock manure pollution, and 1.84 tons/hectare of arable land of livestock manure pollution (Qiu et al., 2013). Livestock manure discharged into the environment without appropriate processing is a serious source of pollution in soil and water systems (Tian, 2012; An et al., 2017).

Our sensitivity analysis found that the model we used to calculate oocyst and cyst emissions was most sensitive to oocyst and cyst excretion rates, similar to the results from the GloWPa-Crypto L1 model (Vermeulen et al., 2017). More detailed information on the toll of cryptosporidiosis and giardiasis and the excretion rates of infected people in Chongqing would improve the model. Additionally, our sensitivity analysis highlighted the significance of runoff. Liu et al. (2019) found that the combined effect of fertilization and runoff played a very important role in oocyst concentration in rivers. Future studies should consider the effect of runoff along with the timing of fertilization. The model was also sensitive to wastewater treatment and manure management. Scenario 3 proposed what would happen if population, urbanization, and sanitation changed similarly to Scenario 1, but without advancements in wastewater treatment and manure management. The results of Scenario 3 indicated that improving urbanization and sanitation with the same population could still result in an increase in surface water emissions if the sewage and manure management systems are inadequate. The analyses of Scenarios 1 and 2 showed a decrease in oocyst and cyst emissions when there were significant improvements in urbanization, sanitation, wastewater treatment, and manure management, along with appropriate population growth. The effects of population, urbanization, sanitation, manure management, and wastewater treatment on oocysts and cysts should be studied in more detail in order to reduce emissions.

Previous studies have used the GloWPa-Crypto model to estimate human and livestock Cryptosporidium emissions across many countries (Hofstra et al., 2013; Hofstra & Vermeulen, 2016; Vermeulen et al., 2017; Vermeulen et al., 2019), but none of these studies included Giardia. In our study, we used the GloWPa-TGR-Crypto model to estimate Cryptosporidium spp. and G. duodenalis emissions from humans and animals in the Chongqing area of the TGR. Unfortunately, the Giardia emissions from rabbits, sheep, and goats were not estimated because there is currently no data for their excretion rates. Earlier studies could not detect Giardia in rabbits, sheep, or goats (Ferguson et al., 2007). Giardia was recently found in sheep and rabbits in northwest and central China, but the cyst excretion rates per kg of manure were indeterminate (Wang et al., 2016; Jin et al., 2017; Jian et al., 2018; Jiang et al., 2018) and may have been underestimated.

To our knowledge, the GloWPa-TGR-Crypto model cannot be validated through the direct comparison of measured surface water values because this method ignores certain factors, such as the infiltration pathways and transport via soils and shallow groundwater to surface water (Bogena et al., 2005; Vermeulen et al., 2017; Watson et al., 2018), the overflow of sewage treatment plants during the flood period (Xiao et al., 2017), traditional dispersive small-scale peasant production (Li et al., 2016), and the excretion of wildlife (Atwill, Phillips & Rulofson, 2003). The pathogen loading data of these factors are not readily available. Despite its few shortcomings, we used the GloWPa-TGR-Crypto model to further study environmental pathways, emissions in the TGR, and sources and scenarios for improved management.

Conclusions

This study is the first to explore Cryptosporidium spp. and G. duodenalis spatial emissions from human and livestock feces in the TGR, and to identify main sources of this pollution. There was a large amount of total emissions in Chongqing (1.6 × 1015 oocysts/year and 2.1 × 1015 cysts/year, respectively), indicating the need for effective pollution countermeasures. The total point source emissions from wastewater containing human excretion in urban areas were greater than the total nonpoint source emissions from human and livestock production in rural areas by a factor of 2.0 for oocysts and 3.9 for cysts. The emissions from urban areas were mainly from domestic wastewater in densely populated areas, while rural emissions were mainly from livestock feces in concetrated animal production areas. Sewage from cities and livestock feces from rural areas are therefore of particular concern in the TGR area.

The GloWPa-TGR-Crypto model was most sensitive to oocyst and cyst excretion rates, followed by prevalence and runoff. If there are significant population, urbanization, and sanitation management changes by 2050, the total Cryptosporidium and Giardia emissions in the TGR will decrease by 42% according to Scenario 1, increase by 15% in Scenario 2, or increase by 28% in Scenario 3. Our scenario analyses shows that changes in population, urbanization, sanitation, wastewater management, and manure treatment should be taken into account when trying to improve water quality. The GloWPa-TGR-Crypto model can be further refined by including direct rural resident emissions, direct animal emissions, emissions from sub-surface runoffs, and a more in-depth calculation of concentrations and human health risks using a hydrological model and scenario analysis. Our model can contribute to further understanding of environmental pathways, the risks of Cryptosporidium and Giardia pollution, and the design of effective prevention and control strategies that can reduce the outbreak of waterborne diseases in the TGR and other similar watersheds.

Supplemental Information

File S1 Supplemental Tables

Click here for additional data file.

Dataset S1 Raw code run in the Matrix Laboratory (MATLAB R2016a) to preparation for Figures 5, 6, 8 and 9

Click here for additional data file.

Dataset S2 Rawcode run in the Matrix Laboratory (MATLAB R2016a) to preparation for Figures 8 and 9

Click here for additional data file.

We would like to thank Prof. Jian Wang from Southwest University for their advice and support.

Additional Information and Declarations

Competing Interests

Author Contributions

Data Availability

The authors declare there are no competing interests.

Qian Huang, Ling Yang, Bo Li and Huihui Du conceived and designed the experiments, performed the experiments, analyzed the data, prepared figures and/or tables, authored or reviewed drafts of the paper, and approved the final draft.

Feng Zhao, Lin Han, Qilong Wang and Yunjia Deng conceived and designed the experiments, performed the experiments, analyzed the data, authored or reviewed drafts of the paper, and approved the final draft.

Guosheng Xiao and Dayong Wang conceived and designed the experiments, performed the experiments, authored or reviewed drafts of the paper, and approved the final draft.

The following information was supplied regarding data availability:

The raw measurements are available in the Supplemental Files.

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
