# Peer review of "Cryptosporidium spp. and Giardia duodenalis emissions from humans and animals in the Three Gorges Reservoir in Chongqing, China"

_PeerJ, doi:10.7717/peerj.9985_

## Round 0.1 · original submission · Major Revisions

Dear Dr. Huang and colleagues:

Thanks for submitting your manuscript to PeerJ. I have now received two independent reviews of your work, and as you will see, the reviewers raised some concerns about the research. Despite this, these reviewers are optimistic about your work and the potential impact it will have on research studying Cryptosporidium spp. and Giardia duodenalis mechanisms of dissemination. Thus, I encourage you to revise your manuscript, accordingly, taking into account all of the concerns raised by both reviewers.

Please ensure that the supplemental data set files are easy and straightforward to access. Please state the software needed to open these files.

Please use the name “GloWPa” in reference to the model that you use.

There appear to be many typos and grammatical errors throughout. Please work on this carefully and enlist the help of another reader to improve the writing quality.

The Materials and Methods appear to be missing important information. All statistical methods should be adequately described such that they are repeatable.

Please provide all of the missing references pointed out by the reviewers.

There are many comments by both reviewers that ask for more information on specific issues; please address these.

I look forward to seeing your revision, and thanks again for submitting your work to PeerJ.

Good luck with your revision,

-joe

·

Basic reporting

Good use of scientific literature.
Paper structure and set up is okay.

The use of English could be improved.
I could not open the raw data, unclear which software is needed.

Experimental design

The research is interesting and relevant.

However, the paper is unclear in the description of exactly which data on what scale are used where and why. Insufficient detail to exactly replicate the research.


- Are calculations really done on a 0.5 degree grid basis? Because the supporting information gives the animal and human population data per district. If the calculations are really done on a grid basis, it should be explained how the results were aggregated to districts (figure 5 and 6).
- Furthermore, a 0.5 degree grid is very coarse for a model for an area of this size (the area would only be about 6 x 10 grid cells, according to figure 5 and 6). The original Hofstra et al. model was a global scale model; then a 0.5 degree grid is justifiable, but if more detailed data for a smaller region is available then it is not justifiable in my view, a smaller grid should be chosen.
- It is unclear from the paper and the supporting information where the data on urban / rural populations come from, and how these are spatially used in the model.
- It is unclear from the paper what data were taken from the original Hofstra 2013 study, and what data are local data for China. For some this is in the supporting information, but not all.
- Why was the year 2013 chosen for the study? Should be explained.
- Line 199: first mention of calculations on a grid basis, this should be introduced earlier.
- Line 203: What are ‘ breeding days’ and why is this variable included in the model? This variable was not in the original Hofstra 2013 model that this research is based on.
- Line 208: Is all manure treated in China? Because this is not standard practice in other places around the world, some more information on this would be helpful.
- Line 230-235: the authors have not thought of this approach themselves, but taken this approach from Hofstra et al. 2013, this should be credited.
- Line 399: what does ‘disposed harmlessly’ mean? This assumption should be explained (in the methods section)
- Line 399-400: What assumption did you make for your calculations about the percentage of manure applied to land? Should be explained in methods section
- Line 408: did you use this number of 1.84 tons in your calculation? If yes, should be explained in your methods section. If not, why not, and what number did you use?
- Line 429 – 431: But if these four studies provide data on Giardia in sheep and rabbits in China, and in fact you provide these data in Table S6, why were these data not incorporated in your model (0 in Table S4)??

- Table S1: the livestock data should be given for the individual animal species instead of livestock total. Which are presumably the different animal species added up? Without the separated data, this research is not reproducible.
- Table S1: Where do these data come from? Reference?
- Table S1: The human population density and livestock density data in Table S1 are given per district. However, the original model by Hofstra et al. 2013 used gridded human and livestock population density data (on a 0.5 x 0.5 degree grid). It is unclear from the paper whether the calculations were done on a grid basis or district basis. The results are only shown in maps on a district basis, but the text does mention grids here and there (e.g. line 199 mentions explicitly that calculations are done per grid cell). As is, this research is not reproducible.
- What are the units of Table S2 in the supporting information?

Validity of the findings

More discussion regarding impact and novelty would be welcome.
Results and discussion are unclear or erroneous in places.


- The results are presented in the text with four significant digits. This suggests an unrealistic amount of certainty that is not justifiable with this modelling approach.
- The population that is classified as non-source means a population that is connected to a sanitation type that produces no emissions to the environment, in the original Hofstra model. However, the text (lines 257 – 269) suggests that emissions are calculated for non-sources, this is very confusing.
- Line 280: are these (oo)cysts per grid cell, per year, per district? No unit given.
- Line 301-302: ‘the data were limited by subjective and objective factors’. Unclear what is meant here.
- Interpretation of what the outcome of the sensitivity analysis means is lacking
- Line 321-322: why is it that people in these regions obtain this treatment? Explanation and interpretation of the patterns in figure 8 is lacking.
- Line 383: This statement that the model generally produces estimates around 1.5-2 log higher than observations is made about the surface water oocyst concentrations model by Vermeulen et al. 2019, NOT the oocyst emission model by Hofstra et al. 2013. It is therefore incorrect and very misleading to apply this statement to this research!
- Line 388: give numbers on the concentrations of Crypto and Giardia that are found in the WWTPs in this area, that would be informative for the reader
- Line 419-421: misleading, because you based your model on these models, they are not completely ‘different models’
- Line 421-424: the work by Liu et al. 2019 sounds like a very relevant work. Can you discuss the assumptions that Liu et al. made regarding the oocyst production and emission in this watershed, and contrast these with your own assumptions?
- Line 432-437: No, you cannot validate your model with surface water values because you did not calculate surface water values, only emissions. If you wanted to calculate surface water values, you should use your emissions as input for a hydrological model!
- Line 453: you did not include actual runoff data in your model, only an assumption on a runoff fraction. This should be made clear to the reader and discussed in the paper!
- Line 458: not just a routing model, but any hydrological information! A routing model is only a part of hydrological modelling.
- The main conclusions from the scenario analysis should be incorporated in the conclusion section


- Figure 1: reference for these data should be given!

- Figure 2: this figure is adapted form Hofstra et al. 2013. Credit should be given!

- Figure 2: Why is the arrow from Rural to Surface water grey? Because as I understand it, you are calculating this?

- Figure 3: Strange y axis scale, bit difficult to interpret. (Especially the strange jumps from 5x10^13 to 6x10^14 to 1x10^15??)

- Figure 4: Incorrect explanation of non-source, as non-source is not an emission category. This figure actually does not show emission categories, but the fraction of the population having a sanitation type that does or does not produce a certain type of emissions.

Additional comments

Examples where the use of English could be improved are listed here. This list is not exhaustive, it is recommended that the entire text be checked.
Plural vs singular / present – past tense:
- Line 139 showed
- Line 245 human  humans
- Line 214: presented  present
Vague wording or too long sentences:
- line 91-95,
- line 95-99,
- line 260-262
- line 395-397
- line 400-403
- line 410-412
- line 416: ‘it is deserved to’
Incorrect words:
- line 51“ are the largest animal emissions”  “ account for the largest animal emissions”
- line 111 whereas should be removed
- line 115 ‘the’ before China should be removed
- line 116 preponderance is strange
- line 118 lead  leading
- line 289: them  those
- line 316: ‘the’ should be removed
- line 328: ‘also’ should be removed
- line 350: ‘from the increasing pollution’  ‘the’ should be removed
- line 437: ‘in actual’ should be removed.
Other textual comments
- Line 54: how can urbanization be ‘ improved’ ?
- Line 65-66: Strange summary. The sentence suggests that children are examples of immunocompromised people, and that AIDS patients and neonatal animals are examples of malnourished individuals.
- Line 69: Hofstra et al. 2013 is not a primary reference for this statement
- Line 73: not all animals are a reservoir of Crypto and Giardia!
- Line 89: does ‘ host’ here refer to animals, humans or both?
- Line 134-136: Text seems to suggest that the figure shows emissions, but it does not
- Line 140: ‘two pollution sources’  ‘ two types of pollution sources’
- Line 212-213: ‘ range of the parameter that can take’  ‘ range that the parameter can take’
- Line 230: insert ‘of’ after ‘understanding’
- Line 321-323: this is an assumption in a scenario and not a fact, should be made clear
- Line 361: commas around adequately should be removed
- General: inconsistent spelling of feces / faeces

Reviewer 2 ·

Basic reporting

The paper entitled ‘Exploring Cryptosporidium spp. And Giardia duodenalis emissions from humans and animals to the Three Gorges Reservoir in Chongqing, China’ provides a useful approach to better understand the environmental flows of the protozoans in this specific area. The paper is comprehensive as it includes human and livestock loads estimations, a sensitivity and a scenario analysis for the year 2050.

This paper essentially is a local application of the Global Waterborne Pathogen (GloWPa) model. This model is cited throughout the paper (all Vermeulen and Hofstra references). However, to enable comparison with other GloWPa model results after publication, this paper should highlight better that it is mostly based on the GloWPa model. Right now it reads as if the authors have developed their own model.

Although being comprehensive and useful, I do also feel the paper in its current form has significant flaws. It will require quite some work to bring it to the level of a good publication. My main criticism is on the methodology and the discussion. I added the minor and textual points below and have added the methods and discussion comments in the other categories below.

Minor points
Several references are out of place. Make sure that you check them again and only really cite the correct literature. E.g. Hofstra et al 2013 is not the source to cite that diarrhea is the third leading cause of death. There are burden of disease studies or WHO documents reporting this. O’Neill et al 2015 is not the source that discusses the lack of available livestock SSP data (and are they indeed still missing or have they in the mean time become available?).

It is important to explain why the loads are relevant. What you say is that this model can contribute to risk assessment (line 460). However, risk assessment is based on the amount a person ingests. How do the loads that you calculated relate to this?

Line 250-251 For human … of total emission reach the TGR. How do you know? How about decay along the way?

Line 279-280 Total annual animal source … emissions. Why are these animal sources spread over larger areas? I see for both human and animal emissions the full map coloured? I don’t understand.

You need to look at the units throughout the text. E.g. line 285 what are the units? (Oo)cysts per district or grid? Or? The units are in more locations unclear.

Line 403-408 What water pollutants did that Census of pollution sources look at? Would you expect the results be different for protozoans? Also, what does this text mean for your results. Should in your case livestock emissions have been larger than human emissions?

Line 444 you mention that Chongqing is an emission hotspot. How do you know? You only studied this area. How will it compare to other areas?

Figure 1 caption what is ‘sewage disposal’?

Figure 1 figure: now that you added rural and urban to the figure (compare to the figure in Hofstra et al 2013), the figure is inconsistent. Are animals, humans, rural, urban, land, WWTP and surface water really all model components as you mention in the caption? In case you do want to add in urban and rural, consider splitting the box Humans up in two, or add rural population and urban population and remove the box with humans.

Figure 3: why not use a log-unit scale. In the current way, the cattle Crypto emissions can not be quantified.

Figure 4: why did you scale this figure to 100%. Why not immediately show the importance of the individual categories? Also, is the y-axis title correct? Shouldn’t it be loads rather than population?

Figure 7: you show differences between the scenarios. In the text you will need to put these differences into perspective. What does it mean that the emissions are almost halved for Scenario 1? How does halving compare to increased log reductions during treatment or the logs difference in the sensitivity analysis?

Figure 8: The difference figures do not clearly show positive and negative. Make this more obvious.

Language

The paper uses emissions and loads interchangeably. However, in line 136 ‘load’ is defined. I would suggest the authors use load throughout the paper and remove the word emissions.

The English language used in the paper is mostly reasonable. However, still quite a number of mistakes are made and therefore I would recommend checking by a native English author or editor. I have picked up on a few mistakes, but this list certainly is not exhaustive:
Line 51 have the largest instead of are
Line 115 remove the before China
Line 122 estimateS
Line 135 humanS
Line 137 end up IN THE surface water instead of on
Line 141 systems THAT reach the TGR
Line 143 in rural areas that feces enter surface water… rephrase (not sure what you want to say)
Line 145 extensive lit review that only found 3 papers? Change wording of extensive.
Line 153 humanS
Line 155 and diffuse emissions. The emission categories…
Line 164 were also reference to … rephrase
Line 191 animalS
Table S4 should be clear on having infected livestock.
Table S5 and 6 can be merged. Also spell average correctly
Line 212 range of the parameter that can take… rephrase, not sure what you want to say
Line 221 2050 WERE based
Line 227-228 does SSP3 emphasize regional progress? Also in times of war? Not sure…
Line 228 strongly instead of highly
Line 230 understanding OF the
Line 251 Humans were responsible for ….% of the oocyst emissions. Line 253 same comment for the cyst emissions.
Line 256 add manure to subtitle
Line 324 urban residentS ARE responsible
Line 344 urban areas instead of residents
Line 351-352 China is one of the regions with Crypto…
Line 360 important instead of a significance
Line 361 remove commas twice
Line 363 aimed AT
Line 386 densely populated urban areas instead of high population density urban
Line 416 consequence, manure treatment, such as ….. on oocyst should be studied in more detail ….
Line 453 Therefore is out of place. The sentence does not follow logically from the previous sentence.

Experimental design

Methodology

The methodology of the paper should be further elaborated. There are many open ends and unclear interpretations. In addition, in its current form, I feel the paper is not reproducible. Examples:
- Why is the year 2013 chosen as baseline?
- The authors have chosen to use human and livestock loads and then split these up in connected, direct and diffuse loads. It is not in all cases clear what these categories involve. For example, in line 157 the authors mention that faecal waste has been collected and used for irrigation as fertilizer. It is unclear to me which faecal waste is meant here. Additionally, in the explanation of the livestock loads there is no mention of dealing with the different categories differently, but Figure 4 does split them up. It is unclear how this split up has been developed. Finally, the methodology does not discuss non-sources. The results and discussion section do. What is this non-source category?
- Further on the faecal waste. Is the the faecal waste from pit latrines (only latrines, not from septic tanks, or the waste from sewer pipes?) directly used on the land? How about storage in the pits before emptying. Would this influence reduction of pathogens in the faecal waste? To solve this point and the previous one, the paper would really benefit from a TGR area-specific explanation of the sanitation situation and livestock manure management.
- Why is the HDI relevant? You are working in one country with only one HDI?
- From Table S2 it seems like the current sanitation fractions used are the same across the districts. Is that indeed the case? This could make quite the difference to the final results and maps.
- The surface runoff fraction used (0.4) seems to be very high. Ferguson uses 2.5%, Vermeulen et al 2019 use even much lower values (2-8 log stay behind on the land). Motivate this fraction of 0.4.
- Only from equation 4, I realised that some of the manure is treated before it is discharged into the surface water (is that my correct interpretation?). Is that my correct interpretation? What happens to the manure otherwise? Is there any manure storage? If so, would you need to consider losses during storage? Also, I happen to know that direct discharges from farms to the surface water regularly occur in China. Are these included? This is all not clear. Again, paint a picture of the situation in China and then explain which of these are included in in what way.
- It is not clear to me why the breeding days of livestock species are relevant. Why is this included? Motivate.
- The sensitivity analysis requires much more explanation. For example, the connected fraction is doubled. How can you get a connected fraction higher than 1? Additionally, the fractions together add up to one. When you change one, the others should also automatically change. How did you deal with this? Etcetera.
- The scenario analysis requires a lot more motivation. The way it is currently discussed, shows a global scale interpretation of the SSPs. However, you are focusing on China alone. Did anyone already do a higher resolution interpretation of the SSPs for China? I would be surprised if nobody has done that yet. It is important to do a local interpretation of such large-scale scenarios, as they may mean something very different in different parts of the world. So how can you interpret the SSP1 and 3 scenarios for China and the TGR region specifically? Also, motivate better why it is OK to assume that there are no changes in excretion rates and prevalence (wouldn’t a better developed country have lower prevalence?). Finally, you currently assume that the livestock density increases at the same rate as the human population. Is that realistic? Motivate. In this way, all choices will have to be more carefully motivated.
- In line 272, at once it is mentioned that spatially distributed maps are produced. How did that spatial distribution happen? And does a 0.5 x 0.5 lat lon degree resolution make sense in the case of this study, or should the resolution be higher, or can you also plot districts (maybe you have done that already, but that is not clear).
- Are there animals in urban areas? Line 265…?

Validity of the findings

Discussion

The discussion of the paper is weak. Many holes are picked in the own work, but the results are not put in perspective of earlier results or the own sensitivity analysis. The GloWPa model is seen as a different model (line 419), but at the same time, it is seen as the same model (at least, that is what I think?) in line 382-384 (‘the model’ on line 383). Nevertheless, results of both models for the same areas are not compared. Why not? This is an easy first step. You have more spatial detail in your model inputs, so your model is potentially the better model. Right?
There are opportunities for more comparison. E.g. the Xiao et al 2013a paper is mentioned on line 369. Measured concentrations are mentioned. However, how can they be compared to your results? You try to make a calculation and arrive at a load, but it is unclear to me 1. how you get to this load and 2. how I can compare this load to the concentrations of Xiao. That is a missed opportunity.
The discussion is full of texts like ‘it is reasonable that our estimated emissions [] exceeded actual loads of waters in the TGR within an order of magnitude’ (line 382) how do you know it is within the order of magnitude?!), the TGR-WP ‘comprehensively’ (line 425) simulated emissions (how do you know? What tells you this?), ‘Although there are many shortcomings, our TGR-WP model can contribute to understanding’ (line 438-440, what makes you conclude this? You just mentioned quite a number of things that were not included in this model. You need more ammunition to make it convincing). The discussion really requires better comparison with available data and much better argumentation. You did a sensitivity analysis. Use this analysis to put your results into perspective!

Additional comments

See text above.

---

## Round 0.2 · Minor Revisions

Dear Dr. Huang and colleagues:

Thanks for revising your manuscript. The one reviewer willing to re-review is mostly satisfied with your revision (as am I). Great! However, there are a few minor edits to make. Please address these ASAP so we may move towards acceptance of your work.

Please also seriously consider the reviewer’s comment about your model name.

Best,

-joe

·

Basic reporting

I think this is okay now, except for one comment:

I think you should not just refer to your model as ‘the GloWPa-Crypto model’ (as is done in the abstract), as this is the name of the original global scale Cryptosporidium model. You should make clear your model is an application/adaptation of the GloWPa-Crypto model, not suggest it is the same, as you apply it locally and not only for Crypto. Perhaps a derived name, such as ‘GloWPa-TGR-Crypto’. Or your own name with a clear reference, like ‘our TGR-Crypto-Giardia model is based on the GloWPa-Crypto model [ref]’.

Experimental design

- You now mention in your discussion that you do a calculation using a hydrological model (lines 416-420). All calculations should be detailed in the methods section and reported in the results section, not in the discussion!! It is unclear now how you did this calculation, and what hydrological information you used. Applying a hydrological model is not something you can mention in one line in the discussion, this is a whole section of your paper at the very least if you actually do these calculations.
- Vermeulen et al. is not the original reference for a hydrological model.
- Furthermore, this hydrological model is on a 0.5 degree grid scale, while you calculate on a district level. Unclear how you would combine this with grid-based hydrological modelling.
- Furthermore, table S11 suggests you also model decay in the surface water according to Vermeulen et al., if this is a calculation you do, it should also be detailed in the methods and results section.
- I still do not understand what the addition of breeding days in equation 3 means. If you already have the number of animals, their manure production and the amount of (oo)cysts in the manure, then what does it add? What exactly does the number ‘breeding days’ represent? Is that the number of days between litters of offspring? The gestation time? The number of days young live before slaughter? Should be explained.

Validity of the findings

- I do not have the Matlab sofware, so I have not checked the supplementary files.

---

## Round 0.3 · accepted · Accept

Dear Dr. Huang and colleagues:

Thanks for again revising your manuscript. I now believe that your manuscript is suitable for publication. Congratulations! I look forward to seeing this work in print, and I anticipate it being an important resource for research studying Cryptosporidium spp. and Giardia duodenalis mechanisms of dissemination and waterborne diseases in the TGR and other similar watersheds. Thanks again for choosing PeerJ to publish such important work.

Best,

-joe